# Complex of Proline-Specific Peptidases in the Genome and Gut Transcriptomes of Tenebrionidae Insects and Their Role in Gliadin Hydrolysis

**DOI:** 10.3390/ijms24010579

**Published:** 2022-12-29

**Authors:** Valeriia F. Tereshchenkova, Irina Y. Filippova, Irina A. Goptar, Yakov E. Dunaevsky, Mikhail A. Belozersky, Elena N. Elpidina

**Affiliations:** 1Faculty of Chemistry, Lomonosov Moscow State University, Moscow 119991, Russia; 2Izmerov Research Institute of Occupational Health, Moscow 105275, Russia; 3A.N. Belozersky Institute of Physico-Chemical Biology, Lomonosov Moscow State University, Moscow 119991, Russia

**Keywords:** proline-specific peptidases, dipeptidyl peptidase 4, transcriptomic analysis, Tenebrionidae insects, gliadin hydrolysis

## Abstract

A detailed analysis of the complexes of proline-specific peptidases (PSPs) in the midgut transcriptomes of the larvae of agricultural pests *Tenebrio molitor* and *Tribolium castaneum* and in the genome of *T. castaneum* is presented. Analysis of the *T. castaneum* genome revealed 13 PSP sequences from the clans of serine and metal-dependent peptidases, of which 11 sequences were also found in the gut transcriptomes of both tenebrionid species’ larvae. Studies of the localization of PSPs, evaluation of the expression level of their genes in gut transcriptomes, and prediction of the presence of signal peptides determining secretory pathways made it possible to propose a set of peptidases that can directly participate in the hydrolysis of food proteins in the larvae guts. The discovered digestive PSPs of tenebrionids in combination with the post-glutamine cleaving cysteine cathepsins of these insects effectively hydrolyzed gliadins, which are the natural food substrates of the studied pests. Based on the data obtained, a hypothetical scheme for the complete hydrolysis of immunogenic gliadin peptides by *T. molitor* and *T. castaneum* digestive peptidases was proposed. These results show promise regarding the development of a drug based on tenebrionid digestive enzymes for the enzymatic therapy of celiac disease and gluten intolerance.

## 1. Introduction

*Tribolium castaneum* and *Tenebrio molitor* are genetic and biochemical model insects in the family Tenebrionidae. These insects are stored product pests of grain crops and wheat flour, and their main dietary proteins are therefore proline- and glutamine-rich gliadins—the prolamins of wheat and the main component of gluten. Gliadins contain up to 50% glutamine and up to 30% proline residues [1,2]. Proline is structurally unique and the only imino acid among 20 natural proteinogenic amino acids. Proline bonds cannot be hydrolyzed by most peptidases with broad substrate specificity because their unique conformation provides a rigid structure of proteins and peptides that protects them from degradation. In nature, only a special group of peptidases, proline-specific peptidases (PSPs), almost exclusively hydrolyzes bonds formed by proline. Due to the need to digest gliadin-containing food, tenebrionids should possess enzymes capable of hydrolyzing the peptide bonds formed by proline, which thus makes them a good natural enzymatic system to study the broad spectrum of PSPs.

Proteolysis of proline-rich proteins and peptides is also important for providing insects with free proline. This imino acid is an important component of insect metabolism as oxidation of free proline in the mitochondria may provide an important energy source for overwintering larvae or flying adults [3,4]. It has been shown that increased levels of proline correlate with specific environmental conditions in insects, which can be observed in overwintering insects and those with cold tolerance [3,5,6]. Thus, the study of peptidases that participate in the release of free proline from dietary proteins are of great interest. Such enzymes are especially important in the metabolism of insects that feed on proline-rich proteins.

Information about known eukaryotic PSPs has been summarized in several reviews [7,8,9]. However, the complexes of PSPs in distinct species, including insects, have not been described. This group of peptidases includes enzymes that specifically hydrolyze peptide bonds formed (i) by the carboxyl group of proline (prolyl oligopeptidase (POP) (which includes the so-called prolyl endopeptidase (PEP), which actually is POP), dipeptidyl peptidases (DPP) 2, 4, 8, and 9, fibroblast activation protein α (FAP), and prolyl carboxypeptidase (PRCP)) and (ii) by the imino group of proline (prolidase (XPD, Xaa-Pro dipeptidase) and aminopeptidases P (APP) 1, 2, and 3). Most of the representatives of PSPs that have been described up to now are exopeptidases; POP is an endopeptidase, and FAP can show both exo- and endopeptidase activities. The rest of the PSPs include a rather diverse group of different exopeptidases that cleave amino acids from both ends of the polypeptide chain.

Besides the above, there also exist substrates with N-terminal Pro residue, which are not hydrolyzed by any PSP from the described complex. These substrates can be hydrolyzed by peptidases with broad substrate specificity with no preference to Pro residue (post-proline cleaving peptidases with broad substrate specificity, PPCPbs). Thus, N-terminal Pro from a polypeptide chain of any length can be cleaved by leucyl aminopeptidase (LAP). The dipeptide Pro-X can be cleaved by cytosolic non-specific dipeptidase (CND), which hydrolyzes dipeptides of any composition, except Pro-Pro.

The list of PSPs described in the comprehensive collection of exhaustively characterized peptidases [8] includes 19 types of PSPs from all kingdoms of living organisms. Previously, we reviewed 13 representatives of human PSPs [9] from two clans according to the MEROPS classification [10]: serine peptidases and metallopeptidases. POP, DPP, and FAP belonged to the S9 family of serine peptidases: S9A (POP), S9B (DPP 4, 8, 9, FAP), S9X (DPP 6, 10). PRCP and DPP 2 were included in the S28 family of serine peptidases. APP1, APP2, APP3, and XPD belonged to the M24 family of metallopeptidases. For effective release of Pro residues from dietary proteins, each animal species used a specific complex of digestive PSPs, which is a shorter list than the whole list of animal PSPs. We could not find publications devoted to a description of digestive PSP complexes in animals including insects, while inhibition of its members being an effective way to control insect pests. Among the human intestinal PSPs that could participate in digestion, only DPP 4 [11] and XPD [12,13,14,15] activities have been reported in the literature.

In this paper, we present a detailed analysis of our own data about PSPs complexes in the midgut transcriptomes of stored product pests (*T. molitor* and *T. castaneum* larvae), *T. castaneum* genome [16], in comparison with the PSP complex in the human genome [9].

## 2. Results

### 2.1. Search for PSP Sets in Tribolium Castaneum and Tenebrio Molitor

Analysis of the *T. castaneum* genome using all eukaryotic PSPs as a query revealed 13 PSP sequences (Table 1): single sequences for POP, DPP 8, APP3, and XPD; two sequences for DPP 4, PRCP, and APP1; and three sequences for DPP 10 (inactive DPP 4 homolog). The *T. castaneum* genome also contained three sequences homologous to human LAP and one sequence homologous to CND. Most of the found sequences were annotated in the UniProt and NCBI databases, but for the DPP4.2_Tc (hereinafter the name before the dot means the type of peptidase, the number after the dot can occur if there are several variants of sequences of one type of peptidase, and the abbreviation after the underscore refers to the organism: Tc—*T. castaneum*, Tm—*T. molitor*, and Hs—*H. sapiens*), DPP10.2_Tc, and DPP10.3_Tc sequences annotation was refined by us during phylogenetic analysis and multiple alignment with human homologs. Comparison of PSP sets found in the *T. castaneum* genome and *T. castaneum* larval gut transcriptome showed that some of the found PSPs were not transcribed in the gut, namely POP_Tc, DPP10.3_Tc, LAP.2_Tc, and LAP.3_Tc; thus, the *T. castaneum* larval gut transcriptome contained 11 PSP sequences and two PPCPbs sequences.

The set of *T. molitor* PSPs was analyzed using the larval gut transcriptome. It contained 11 predicted protein sequences of PSPs similar to those presented in the *T. castaneum* and human genomes and the *T. castaneum* larval gut transcriptome (Table 1): single sequences for POP, DPP 8, PRCP, APP1, APP2, APP3, and XPD and two sequences for DPP 4 and DPP 10. We also found sequences homologous to LAP and CND.

The sets of human and insect PSPs are presented in Table 1. Overall, the sets of PSPs for all studied organisms were similar, but insects did not possess some of the PSPs found in humans: DPP 9 (which is very close to DPP 8 and DPP 4), FAP, DPP 6 (an inactive homolog of DPP 4), and DPP 2. Interestingly, insects, unlike humans, had some PSPs duplicated (DPP 4, DPP 10, PRCP, APP1, and LAP), and some of the duplicates were found only in the *T. castaneum* genome.

### 2.2. General Characteristics of the PSP Sequences

#### 2.2.1. Serine PSPs

For all peptidases, residues that determine primary specificity were identified. Active sites of active serine PSPs from S9A, S9B, and S28 consisted of a highly conserved catalytic triad in the order Ser, Asp, His. Surrounding residues of all peptidases were also very conservative, with few synonymic substitutions.

Peptidases from the S9A subfamily hydrolyze substrates of the general formula (Xaa)_1-15_-Xbb-Pro↓Xbb-(Xaa)_1-15_, where Xaa represents any amino acid and Xbb represents any amino acid except Pro. The active site of the S9A serine peptidase subfamily including POP consists of Ser554, Asp641, and His680 (hereinafter numbering is given by human POP, UniProt P48147) with conserved sequence motifs NGGSNGG, ADHDDRV, and RAGHGAG [17,18,19]. In the sequences found in Tenebrionidae insects (POP_Tc and POP_Tm), composition of the catalytic triad Ser, Asp, His was similar, with surrounding residues and residues forming binding subsites being highly conserved.

Peptidases from the S9B subfamily hydrolyze substrates of the general formula Xbb-Pro↓Xbb-(Xaa)_n_, where n represents any number. The active site of the S9B serine peptidase subfamily consists of Ser630, Asp708, and His740 (hereinafter numbering is given by human DPP 4, UniProt P27487). Homologous sequences of the S9B serine peptidase subfamily found in Tenebrionidae insects had the same order in the catalytic triad. In the structure of human DPP 4, the Pro (P1) residue of the substrate is located in the hydrophobic S1 subsite surrounded by Tyr631, Val656, Trp659, Tyr662, Tyr666, and Val711. The S2 subsite includes Asn710, Arg125, Glu205, and Glu206 [18,20,21]. Active insect peptidases contained conserved substrate-binding subsite residues, with the only exception being the S1 residue Trp659, which was replaced in DPP4.1_Tc Trp659Leu and DPP4.1_Tm Trp659Phe. All DPP 4-like peptidases found in Tenebrionidae insects, together with human DPP 4, possessed only exopeptidase activity, having Asp or Asn residues in an important Asp663 position similar to human DPP 4. This fact distinguished DPP 4 from FAP, containing in human smaller Ala657 residue [22] in a corresponding position, which explains the presence of endopeptidase activity in addition to exopeptidase activity in FAP [20,22,23,24]. An inactive homolog of DPP 4 (human DPP 10) had a Gly residue instead of Ser630 in the active site [25,26]. In the *T. castaneum* genome, three sequences of DPP 10 were found. Two of them also had a Gly residue instead of Ser630 and formed homologous pairs with *T. molitor* sequences. The third—DPP10.3_Tc—had Ser residue in the active site similar to active members of the S9B subfamily, and presumably can thus have enzymatic activity similar to DPP 4, although the sequence structure is characteristic of DPP 10.

PRCP from the S28 subfamily hydrolyzes substrates of the general formula (Xaa)_n_-Xbb-Pro↓Xbb. DPP 2 hydrolyzes substrates of the same general formula as the S9B subfamily, cleaving Xbb-Pro dipeptide from the N-terminus, as described above; however, no such activity was found in tenebrionids. The catalytic triad of the S28 serine peptidase family contains Ser179, Asp430, and His455 (hereinafter the numbering is given by human PRCP, UniProt P42785) residues. PRCP sequences from Tenebrionidae insects also had conserved residues surrounding the catalytic triad with minor replacements, as well as similar Pro-binding S1 subsites (Table 2).

#### 2.2.2. Metallodependent PSPs

All metallodependent PSPs (APP1, APP2, APP3, and XPD) belong to the M24B subfamily and are double Mn^2+^-dependent enzymes, requiring both ions for activity [27]. APP hydrolyzes substrates of the general formula Xbb↓Pro(Xaa)_1-9_. XPD hydrolyzes dipeptides of the general formula Xbb↓Pro. Insect peptidases contained two conserved metal-binding sites, as well as residues essential for maintaining activity similar to human enzymes (Table 3).

#### 2.2.3. Post-Proline Cleaving Peptidases with Broad Substrate Specificity (PPCPbs)

Additional PPCPbs (LAP and CND) are also metallopeptidases. LAP hydrolyzes substrates of the general formula Xaa↓Xbb-(Xaa)_n_. It is a Zn^2+^-dependent peptidase from the M17 family and contains two Zn^2+^ ions. CND hydrolyzes dipeptides of the general formula Xaa↓Xbb. It is from the M20 family and contains two Mn^2+^ ions. In general, all sequences found in Tenebrionidae insects were highly conserved in metal-binding sites (Table 4, numbering is given by human peptidases).

This analysis showed that all found PSP sequences from *T. molitor* had a conserved structure compared to *T. castaneum* and human enzymes sequences, which made sequence annotation easier.

### 2.3. Comparison and Phylogeny of PSPs in T. molitor and T. castaneum

The phylogenetic tree represented in Figure 1 includes *H. sapiens*, *T. castaneum,* and *T. molitor* PSPs. The tree was divided into separate branches, each corresponding to one of the three families of PSPs (S9, S28, M24) or one of the two PPCPbs (M20 and M17). Each branch in turn was subdivided into types of peptidases, which formed orthologous pairs. A clade of DPP 4 proteins was divided into two distinct orthologous pairs: DPP4.1_Tc–DPP4.1_Tm and DPP4.2_Tc–DPP4.2_Tm. Insect DPP 10 formed a separate clade. Among them, DPP10.1_Tc–DPP10.1_Tm and DPP10.2_Tc–DPP10.2_Tm formed two distinct orthologous pairs. DPP10.3_Tc had a Ser residue in the catalytic triad that was located near the second pair. Separate clades were also formed by other PSPs. Distinct groups were formed by the DPP 8, POP, PRCP, APP1, APP2, APP3, XPD, CND, and LAP proteins. In each clade, homologous insect sequences formed orthologous pairs. Human proteins were located near them due to the high structural and functional similarities.

### 2.4. Localization of PSPs

The predicted molecular masses of serine PSPs from the S9 and S28 families ranged from 54 to 103 kDa, and for metallodependent PSPs from the M24B subfamily these predictions ranged from 54 to 88 kDa. In most cases, calculated molecular masses were similar for enzymes of the same type (Table 5).

To get closer to understanding the possible functions of PSPs, we analyzed the presence of specific localization domains connected with secretory pathways, which in this case was the signal peptide (with transportation to the gut or lysosomes). We also searched for the mitochondrial transit peptide and transmembrane domains. The results of this study are summarized in Table 5.

Human and insect POP was predicted to lack signal peptide according to SignalP-6.0. However, POP_Tm was predicted to have it by the TargetP-2.0 server. Based on previous studies, human [28] and *T. molitor* [29] POP were shown to be cytoplasmic enzymes.

Among DPP 4 sequences, DPP4_Hs had a small transmembrane domain (20 amino acid residues). The TargetP-2.0 server predicted signal peptides for this sequence, but the probability of its appearance was not very high. According to experimental data, this protein exists in two forms: membrane-bound with a large extracellular domain on the cell surface [30] or soluble protein resulting from the cleavage of the extracellular domain from the short transmembrane part [31]. Tenebrionidae insects had two pairs of sequences coding DPP 4 proteins. The first homologous pair of sequences (DPP4.1_Tc–DPP4.1_Tm) had signal peptides predicted with high probability and proteins were presumed to be secreted. The other pair of DPP 4 sequences (DPP4.2_Tc–DPP4.2_Tm) had transmembrane domains and no signal peptide, similar to the human protein. The rest of the DPP 4-like proteins (DPP 8 and DPP 10 for both humans and insects) had no signal peptides. Human DPP 8 was shown to be a cytoplasmic peptidase [32,33,34]. All DPP 10 sequences had transmembrane domains, which is supported by previous data for this human protein [25,26,35].

Both human and insect PRCP sequences from the S28 family had signal peptides and were predicted to be secreted in agreement with previous studies [29,36,37,38,39,40].

Almost all metallopeptidase sequences had no specific domains connected to secretion, except for human and *T. molitor* APP2 and *T. castaneum* APP1.2_Tc, which had N-terminal signal peptides predicted. The APP1.2_Tc sequence is currently annotated in UniProt and NCBI databases as APP 1-like protein, but due to the predicted localization domain and conducted phylogenetic analysis (Figure 1) it probably belongs to APP2 proteins. Although there are no reliable data on the localization of such peptidases in the studied insect, there is some information about the localization of mammalian peptidases. It is known that signal peptide can serve as a signal-anchor sequence for transmembrane proteins of type III [41] and this is supported by data for human APP2, which was shown to be plasma membrane peptidase [42]. APP1 was localized predominantly in the cytoplasm, and this is supported by data for rat APP1 [43]. Human APP3 had mitochondrial localization [44], which is consistent with the predicted presence of the mitochondrial transit peptide in APP3_Hs and APP3_Tc (high probability), as well as its predicted presence in APP3_Tm (low probability). All LAP sequences, except for LAP.1_Tc, had the predicted mitochondrial transit peptide. Human LAP was found predominantly in microsomal fractions and cytoplasm [45], and CND was found also in cytoplasm with ubiquitous tissue distribution [46].

Overall, there was good agreement between the localization of predicted bioinformatics and experimental data on mammalian PSPs found in the literature (Table 5). Based on the predicted signal peptides for insect peptidases, only DPP4.1 and PRCP were presumed to be secreted proteins.

These bioinformatics predictions correlate with our previous and new biochemical data on the localization of several PSPs in the midgut of *T. molitor* larvae. We found that DPP 4 [48] and PRCP [29] were located mainly in the AM (anterior mesenteron, anterior part of the midgut), POP [29] was distributed evenly between the AM and PM (posterior mesenteron, posterior part of the midgut), and XPD [49] was localized predominantly in the PM. DPP 4 and PRCP were located in soluble gut contents, while POP and XPD were mainly located in tissue fractions.

We also show that the remaining PSP (APP) is distributed evenly between AM and PM parts of the gut, with more than half of its total activity belonging to soluble tissue fractions (Table 6).

### 2.5. Analysis of the Expression Levels of Genes Coding PSPs in T. castaneum and T. molitor Guts

Data on expression levels of genes coding the identified *T. castaneum* and *T. molitor* gut PSPs are presented in Table 7 as reads per kilobase per million mapped reads (RPKM). For better comparison of the possible digestive role of PSPs between two insects, we also calculated the expression levels as a percentage from the sum of expressions of all active PSPs. This calculation involved only active PSPs: POP, DPP 4, DPP 8, PRCP, APP1, APP2, APP3, and XPD. DPP 10 was not included in this calculation because it is an inactive DPP 4 homolog and does not possess peptidase activity, and LAP and CND were not included because they possess broad substrate specificity and the cleavage of post-proline bonds is only additional, not primary.

The highest expression level of S9 PSP mRNA was observed in the orthologous pair of secreted DPP4.1_Tc–DPP4.1_Tm (25% each from the sum of expressions of all active PSPs). The membrane pair DPP4.2_Tc–DPP4.2_Tm had lower expression levels of 8% and 17%, respectively.

The expression level of mRNA in *T. castaneum* PRCP (PRCP.1_Tc) was only 0.4%, whereas orthologous mRNA in *T. molitor* PRCP (PRCP_Tm) exhibited a 9% expression level. However, *T. castaneum* had another sequence (PRCP.2_Tc) with an expression level of 10%, for which there was no ortholog in the *T. molitor* transcriptome.

Among metal-dependent PSP mRNA, the APP1.1_Tc–APP1_Tm pair possessed high expression levels of about 36% and 15%. High expression was also evaluated for the XPD_Tc–XPD_Tm pair, about 19% and 25%, respectively.

All other mRNA of active PSPs from both organisms had expression levels of less than 5% of the sum of expressions of all active PSPs, including the single oligopeptidase POP that was expressed only in the gut of *T. molitor* (POP_Tm, 4%).

Among the rest of the mRNA sequences, there was a highly expressed inactive DPP 4 homolog in *T. molitor* (DPP10.1_Tm) whereas both orthologs of *T. castaneum* were low in terms of expression. PPCPbs mRNA (LAP and CND) had comparable expression levels in both insects, and CND had the highest expression levels among studied sequences in *T. molitor* and *T. castaneum*.

In general, there were similar expression profiles of PSP mRNA in both *T. castaneum* and *T. molitor* (Figure 2). Prospective orthologs had comparable expression levels.

The highest level of gut expression among active PSPs belonged to the exopeptidases DPP 4, PRCP, APP1, and XPD, which may indicate their important role in tenebrionid physiology, particularly participation in the digestive process. It should be noted that POP had a low or non-existent level of gut expression, which makes its participation in the proteolysis of food proteins unlikely.

Localization studies together with the evaluation of the expression level of genes coding *T. molitor* PSPs in the gut transcriptome, as well as the presence of signal peptides connected with secretion, allowed for confirmation of the set of peptidases that can directly participate in the hydrolysis of food proteins in insect larval guts. This set consists of DPP4.1_Tm and PRCP_Tm, which are characterized as secreted peptidases localized in soluble gut contents with high expression levels of their mRNA in the gut transcriptome.

### 2.6. Gliadin Hydrolysis

Gliadin, a major storage protein of wheat, cannot be completely hydrolyzed by human digestive peptidases with the usual broad specificity. Proline- and glutamine-rich peptides, with the high content of proline and glutamine residues in their composition [1,2], remain unhydrolyzed [50] and can cause the development of diseases of the human gastrointestinal tract in predisposed people. In this regard, digestive peptidases of Tenebrionidae insects, in particular PSPs, may present the potential to hydrolyze these remaining fragments by serving as a proposed oral medicine as tenebrionids are stored product pests and gliadins are their major dietary proteins.

We studied the effect of the major luminal digestive *T. molitor* peptidases DPP4.1_Tm and PRCP_Tm, which are capable of cleaving bonds formed by proline on the total preparation of wheat gliadins (Figure 3). In addition, the effect of both individual PSPs and their combination with digestive post-glutamine cleaving peptidases of *T. molitor* (the total preparation of larval digestive cysteine cathepsins (CC)) [51] was studied. Proline-specific luminal enzymes (DPP 4 and PRCP) are exopeptidases, and CC are endopeptidases. Thus, we were able to study the combined action of endo- and exopeptidases on gliadins.

The enzymes were taken in equal amounts per larvae. Among the individual enzymes, the highest intensity of gliadin hydrolysis was achieved by the action of post-glutamine cleaving endopeptidases: CC preparation (62% of which belongs to the major digestive cathepsin L of *T. molitor* NCBI ID AJF94885 according to [52]) was eight times more active than studied PSPs. Among the used PSPs, DPP 4 exhibited 1.5-fold higher activity with gliadins than PRCP. Separate addition of DPP 4 and PRCP to CC increased hydrolysis by 1.4 and 1.3 times, respectively. The combination of all three peptidase preparations improved the rate of hydrolysis by 1.7 times compared to single CC preparation (Figure 3). It should be noted that, in all cases, the combined effect of endo- and exopeptidases was greater than the sum of the separate enzyme actions. This indicates that hydrolysis with endopeptidases forms additional substrates for exopeptidases, and this leads to an increase in the hydrolysis rate of the original protein.

### 2.7. Hypothetical Scheme for the Complete Hydrolysis of Glutamine- and Proline-Rich 26-Mer Immunogenic Gliadin Fragments by T. molitor and T. castaneum Digestive Peptidases

Based on the obtained data, we suggested a hypothetical scheme for the complete hydrolysis of some gliadin fragments by *T. molitor* and *T. castaneum* digestive peptidases (Figure 4). As an example, we have taken the 26-mer FLQPQQPFPQQPQQPYPQQPQQPFPQ of γ-gliadin (UniProt: Q8L6B2), which along with the 33-mer peptide is a strong immunogenic peptide [50]. During the first stage of proteolysis, digestive post-glutamine cleaving endopeptidase CC can most effectively hydrolyze peptide bonds between two glutamine residues. Despite the ability of CC to hydrolyze bonds after the glutamine residue [53], the QP bond should not be cleaved by them, since the bonds formed by proline residue are hydrolyzed by the PSPs almost exclusively. Resulting fragments can be the substrates for DPP 4, which cleaves dipeptides such as QP, FP, and YP from their N-terminus. The remaining tripeptides (QPQ) can be effectively hydrolyzed by PRCP. Prolidase XPD may complete gliadin degradation, cleaving final dipeptides to single amino acids in the cytoplasm of the epithelial cells in the gut [49]. The remaining fragment FLQP is not immunogenic [50].

According to the scheme, the cleavage of bonds formed by glutamine residues by CC is an important first step in the hydrolysis process, and each of the found PSPs is essential for complete digestion of glutamine- and proline-rich gliadin fragments in the midgut of tenebrionids.

## 3. Discussion

A bioinformatic search for PSPs in the *T. castaneum* genome and both Tenebrionidae larvae gut transcriptomes revealed 14 distinct types of sequences in both insects: one POP, two DPP 4, one DPP 8, two PRCP, two APP1, one APP2, one APP3, one XPD sequence (all active peptidases), and three DPP 10 (inactive homologs of DPP 4). There also were two types of peptidases found with broad substrate specificity, possessing the ability to cleave bonds formed by proline residue (three LAP and one CND). The main difference in PSP sets in Tenebrionidae larvae gut transcriptomes was that *T. castaneum*, unlike *T. molitor*, had two PRCP sequences, two APP1 sequences, and lacked APP2 and POP sequences. Generally, Tenebrionidae insect PSP sets were similar to those in humans, lacking only DPP 9, FAP, DPP 2, as well as DPP 6, which is inactive homolog of DPP 4 together with DPP 10 found in all three species. Inactive homologs are represented in all major peptidase families. According to the MEROPS database, it is the serine peptidases that have the highest number of identified inactive enzymes, especially in insects. It has been suggested that a large number of both inactive and active serine peptidases indicates that the evolution of inactive peptidases and their new functions were beneficial for insects [54]. We have also shown a large number of serine peptidases and their inactive homologs in *T. molitor* [55], which can indicate that the emergence of inactive homologs is not an accident, but rather a natural process.

Phylogenetic analysis and multiple alignment of Tenebrionidae insect PSPs with those of humans allowed us to annotate all *T. molitor* peptidase sequences and refine annotation for several *T. castaneum* peptidase sequences based on analysis of cladograms of the tree, active centers, and substrate binding subsites.

The major digestive organ of *T. molitor* larvae is the midgut, where pH varies from 5.2 to 6.5 in the AM and 7.8 to 8.2 in the PM [56,57,58]. This gradient determines localization of the activity of different digestive enzymes along the midgut in accordance with their pH optima [59]. Gut peptidases may be localized either inside the gut lumen contents, and thus directly participate in digestion, or in the epithelial tissues lining the gut lumen. The combination of gene expression studies in the larval gut transcriptome, bioinformatic prediction of signal peptides, and biochemical localization experiments demonstrate the digestive function of the highly expressed secreted PSPs.

Using our previous and new data on the localization of *T. molitor* PSPs, we can summarize that DPP 4 [48] and PRCP [29] were located mainly in the AM, POP [29] and newly studied APP were distributed evenly between the AM and PM parts of the gut, and XPD [49] was localized predominantly in the PM. Analysis of the tissue distribution of the PSPs showed that DPP 4 [48] and PRCP [29] were located in soluble gut contents and can participate in *T. molitor* digestion, and this is in agreement with bioinformatic data (these peptidases were predicted to be secreted and highly expressed). The remaining POP, APP, and XPD were localized in tissues. Among them, XPD definitely seems to participate in digestion during the final stages as the substrates for XPD are dipeptides that can be absorbed by epithelial tissues [49]. There is also the possibility of APP participating in digestion because it can cleave oligopeptides and tripeptides, and tripeptides also can be absorbed by midgut epithelia [60]. POP hydrolyzes only oligopeptides, which cannot be absorbed by tissues. Compared to POP and APP, the participation of XPD in gliadin hydrolysis seems most likely and is supported by its high expression level. Since about 60% of XPD activity was located in the PM, it can act during the final stages of digestion and thus provide complete hydrolysis of gliadins.

Information from various sources about PSPs that can effectively break down gliadins was summarized in a previous review [61]. Attempts to hydrolyze gliadins were mostly limited by the search for peptidases capable of cleaving only peptide bonds formed by proline, such as prolyl oligopeptidases from *Myxococcus xanthus*, *Sphingomonas capsulata*, and *Flavobacterium meningosepticum* [62], prolyl endopeptidase from *Aspergillus niger* [63], and PSPs from the lactic acid bacteria *Lactobacillus plantarum* and *Pediococcus pentosaceus* [64]. However, the use of single PSPs did not achieve the complete hydrolysis of gliadins, and only the discovery of cysteine peptidase from *Hordeum vulgare* (EP-B2) in 2006 [65,66] allowed for the creation and testing of a dosage form for celiac disease treatment, which was based on a mixture of prolyl oligopeptidase from *S. capsulata* and cysteine peptidase from *H. vulgare* (EP-B2) (ALV003 Alvine, USA) and which is now under clinical trial [67,68]. The other cysteine peptidase capable of participating in the hydrolysis of gliadins was later described from *Triticum aestivum* [69].

Previously, we showed that *T. molitor* DPP 4 is able to hydrolyze gliadins more efficiently than human DPP 4 [48]. In the present study, we tested a natural complex of digestive enzymes of tenebrionids to discover whether this would increase the efficiency of hydrolysis by providing efficient hydrolysis of their proline-rich natural food substrates (such as gliadins) in natural conditions and ensuring sufficient intake of proline. Gliadin hydrolysis was studied using single digestive *T. molitor* peptidases and their mixtures capable of cleaving bonds formed by glutamine and proline residues. Among the separate enzymes, the highest intensity of hydrolysis was achieved by the action of post-glutamine cleaving cysteine cathepsins (CC), which corresponds with the hypothesis that they arose during evolution for the efficient hydrolysis of gliadins [51]. Individual digestive PSPs (DPP 4 and PRCP) were less active on intact gliadins, but the combined action of endo- and exopeptidases demonstrated the synergistic effect; this indicates that, after hydrolysis by endopeptidases, additional substrates for exopeptidases were formed, thus leading to an increase in the hydrolysis rate of the original protein.

Based on the obtained data, we suggested a hypothetical scheme for the complete hydrolysis of glutamine- and proline-rich 26-mer immunogenic gliadin fragments by *T. molitor* and *T. castaneum* digestive peptidases. The first stage of hydrolysis was carried out with endopeptidase CC, which can hydrolyze bonds formed by glutamine residues. DPP 4 then cleaved dipeptides from the resulting oligopeptides. The remaining tripeptides were hydrolyzed by PRCP. During the final stage, XPD degraded dipeptides to single amino acids in the cytoplasm of midgut epithelial cells.

The suggested hypothetical scheme is original and interesting because it can explain the hydrolysis of glutamine- and proline-rich gliadin fragments from different gliadins, including not only γ-gliadins (UniProt: P21292) containing PQQPFPQ repeats and ω-gliadins (UniProt: Q9FUW7) with PQQPFPQQ (which are the richest in glutamine and proline residues) but also α-gliadins (UniProt: P02863) with PQPQPFP and PQQPY repeats. Importantly, considering the suggested scheme, several proline- and glutamine-rich gliadin peptides may be degraded, such as QLQPFPQPQLPY, PQPQLPYPQPQLP, LGQQQPFPPQQPYPQPQPF, LQLQPF(PQPQLPY)_3_PQPQPF, and FLQPQQPF(PQQ)_2_PY(PQQ)_2_PFPQ that are not hydrolyzed by human digestive peptidases [50,70] and which cause autoimmune celiac disease. This indicates that it may be possible to develop an enzyme therapy drug for celiac disease and gluten intolerance that is based on tenebrionid digestive enzymes.

## 4. Materials and Methods

### 4.1. Preparation of the Biological Material and Sequencing of cDNA

Preparation of biological material from *T. molitor* and *T. castaneum* larvae guts and sequencing of cDNA was described previously [52]. The preparation of total gut RNA from *T. molitor* larvae from three independent biological replicates was sent to a sequencing facility (National Center for Genome Resources (NCGR), Santa Fe, NM, USA), where mRNA was isolated by polyA, standard libraries were made, and paired-end sequencing was performed on an Illumina HiSeq 2000 (San Diego, CA, USA) using standard protocols from the manufacturer. We obtained approximately 240 million sequence reads with a 250 bp insert.

Paired-end sequencing on the Illumina HiSeq 2000 of *T. castaneum* cDNA was performed by the High Throughput Genomics Center, Seattle, USA. We obtained approximately 340 million sequence reads, and insert size was approximately 250 bp.

### 4.2. Assembly of Contigs

A custom assembly of *T. molitor* sequences combined from all replicates was made by the NCGR, resulting in 197,800 contigs (minimum length = 100 and maximum length = 51328; Q1 = 123, Q2 = 153, Q3 = 335; N50 = 2232, B1000 = 71.9%, B2000 = 54.1%). For *T. molitor* sequences, we also combined the replicate data and included previous Sanger sequencing [71] and pyrosequencing [72] databases of mRNA from the larval gut and performed additional de novo assemblies using SeqManNGen (v. 4.0.1.4, DNAStar, Madison, WI USA) and custom assembly programs. For *T. castaneum* sequences, we used SeqManNGen to map sequences to the *T. castaneum* genome (Tcas3, NCBI; parameters for alignment were merSize = 19; 309,572,610 bp submitted, 263,305,494 aligned, 17,268,742 unaligned; sequence count score was >90%), as well as the Galaxy platform [73,74]. Potential coding sequences, starting at methionine and covering at least 20% of the mRNA sequence, were found in the *T. molitor* contigs using custom software.

### 4.3. Search for Insect PSP Sequences

Reference sequences were taken from the human genome and downloaded from the UniProt database [75,76]. The list of reference human PSPs is given in Table 1. Initially, translated *T. molitor* ORFs homologous to human PSP query sequences were determined using the algorithm tblastx [77,78]. Multiple alignment was constructed by means of pairwise alignments using the Wagner–Fisher algorithm [79]. ORFs that were grouped into blocks with an identity of at least 95% and that overlapped with another block of at least 10 amino acid residues were considered as referring to one unique peptidase. Multiple sequence alignment (MAFFT) [80] was used to refine and build consensus sequences. The predicted start of the *T. molitor* mature enzyme sequences was identified by sequence homology through alignment with mature human PSPs. Multiple alignments of the analyzed amino acid sequences were performed using the program Clustal Omega [81,82].

*T. castaneum* peptidase sequences were found in the NCBI database [83,84] via Protein BLAST, using human PSPs as queries. Found sequences that lacked annotation in the database (namely DPP4.2_Tc, DPP10.2_Tc, and DPP10.3_Tc) were annotated during the subsequent construction of a phylogenetic tree and the analysis of multiple alignments.

### 4.4. Phylogenetic Analysis and Orthology Predictions

A phylogenetic tree of human, *T. castaneum,* and *T. molitor* PSPs was constructed in MEGA X [85,86] using the maximum likelihood method and 200 bootstrap repetitions. Based on the cladogram, pairs of sequences from *T. castaneum* and *T. molitor* were considered orthologous if they formed a single clade.

### 4.5. Analysis of the Expression Levels

Analysis of the expression levels was carried out as written in [52]. To analyze the expression of peptidase transcripts in *T. castaneum*, we used reads mapped to the genome of *T. castaneum* (Tcas3) by SeqManPro (DNAStar) to obtain expression values in normalized reads per kilobase per million mapped reads (RPKM) [87].

For *T. molitor*, about 40 million reads were obtained from each of the three RNA-Seq datasets. BLAST was used to obtain all nucleotide sequences in the contigs that potentially encode peptidases. The contigs were used to assemble and refine sequences of complete peptidase mRNA. Refined peptidase mRNA sequences were used to identify contigs in each assembly with at least 97% sequence identity, which were then used for expression calculations. If a contig aligned only partially to the mRNA, its contribution to expression was proportional to the aligned part if it was more than 50% of the contig’s length. RPKM was used to calculate the number of reads mapped to a contig when calculating each multiread as one unit.

### 4.6. Analysis of the Structure of PSP Sequences

Active centers and substrate binding subsites were analyzed using multiple alignments of studied peptidases (performed in Clustal Omega [81,82] and visualized in GeneDoc [88,89]). The SignalP-6.0 server [90,91] and TargetP-2.0 [92,93] were used to analyze N-terminal parts in the translated mRNA sequences, predict the presence and location of signal peptide cleavage sites and the subsequent secretion of proteins, and predict the presence of mitochondrial transit peptides. Trans-membrane domains were predicted using the DeepTMHMM server [94,95]. Molecular masses were calculated using the online server ExPASy [96]. Molecular masses and sequence lengths for human and *T. castaneum* PSPs were taken from the UniProt and NCBI databases.

### 4.7. Enzyme Assays Using Chromogenic p-Nitroanilide Substrates

Peptidase activity with chromogenic *p*-nitroanilide substrates was determined spectrophotometrically according to [97] at 405 nm with an ELx808 Absorbance Microplate Reader (BioTek Instruments, Inc., Winooski, VT, USA) and was calculated using the initial rates of hydrolysis.

Exact protocols for studied peptidases were described earlier: CC with 0.5 mM Glp-Phe-Gln-pNA (was obtained according to [53]) at pH 5.6 with 6 mM Cys [53], DPP 4 with 0.25 mM Ala-Pro-pNA (was synthesized according to standard procedures [98]) at pH 7.9 [48], and PRCP with 0.5 mM Z-Ala-Ala-Pro-pNA (was synthesized according to standard procedures [98]) at pH 5.6 [99].

### 4.8. Enzyme Assays Using Fluorogenic Substrates

The reaction mixture was placed in a microplate well and included 5-50 µL of APP enzyme, 10 µL of 15 mM MnCl_2_, and 50 mM Tris-HCl buffer (pH 7.5) to a total volume of 195 µL. After 20 min of preincubation at room temperature, 5 µL of 3 mM Lys(Abz)-Pro-Pro-pNA (Bachem, Bubendorf BL, Switzerland) dissolved in DMF was added. APP activity toward the fluorogenic substrate Lys(Abz)-Pro-Pro-pNA was determined at λ_ex_ 340 nm and λ_em_ 400 nm with a FLx800 Fluorescence Microplate Reader (BioTek Instruments, Inc., USA), and was calculated as described in the previous section.

### 4.9. Localization Studies

For localization studies, fractionation of AM (anterior mesenteron, anterior part of the midgut) and PM (posterior mesenteron, posterior part of the midgut) parts from *T. molitor* larvae was performed as described in [48]. AM and PM sections were isolated from fourth-instar *T. molitor* larvae and were split longitudinally. The contents were extruded and retained, while the tissue was rinsed with 0.9% saline. The lumen contents or the gut tissues were pooled from 50 AM or PM (for each of the four biological replicates) and were homogenized separately at 4 °C in 0.5 mL of double distilled water in a glass Dounce homogenizer and centrifuged for 1 h at 35,000× *g*. There were two fractions after centrifugation of the lumen contents: supernatant (soluble contents) and precipitate (insoluble contents). After centrifugation of the tissues there were also two fractions: supernatant (soluble tissues fraction) and precipitate (insoluble tissues fraction). All fractions were stored at −70 °C. The activity of the studied peptidase was measured in each fraction using the same larval gut equivalents of material and the corresponding substrate. Measurements were carried out in 3 biological replicates with 3–5 technical repetitions. Standard deviation did not exceed 5–10%. The relative content of each peptidase in the fractions was then calculated.

### 4.10. Gliadin Proteolysis Assay

To isolate digestive peptidases, extract from 200 pooled AM gut sections (2–2.5 mL) was applied to a Sephadex G-150 column (2.7 × 110 cm). Elution was performed with 0.01 M phosphate buffer (pH 5.6) containing 0.5 M NaCl at 4 °C. Fractions of 9.0 mL in volume were collected and analyzed for protein content at 280 nm and for hydrolytic activity with a specific chromogenic substrate for each peptidase: 0.5 mM Glp-Phe-Gln-pNA at pH 5.6 with 6 mM Cys for CC, 0.25 mM Ala-Pro-pNA at pH 7.9 for DPP 4, and 0.5 mM Z-Ala-Ala-Pro-pNA at pH 5.6 for PRCP. Active fractions corresponding to the major peak of each peptidase’s activity were pooled separately, concentrated, and desalted on Amicon YM3 membranes (Amicon, Winsum, Netherlands) at 4 °C and either used immediately for further analysis or stored at −70 °C.

Proteolysis of gliadins was measured by an increase in free amino groups with 2,4,6-trinitrobenzene sulfonic acid (TNBS) (Reachem, Moscow, Russia) [100]. For a comparative assay of activity, the enzymes were taken in equal amounts per larvae. We used an aliquot of studied peptidases (CC, DPP 4, or PRCP) corresponding to the activity of 1.7 *T. molitor* larval guts. This was calculated using data regarding the activity of the enzymes in the initial midgut extract in the presence of 2 M urea. The activity of CC was measured with 0.5 mM Glp-Phe-Gln-pNA at pH 5.6 with 6 mM Cys, DPP 4 was measured with 0.25 mM Ala-Pro-pNA at pH 7.9, and PRCP was measured with 0.5 mM Z-Ala-Ala-Pro-pNA at pH 5.6. The gliadin proteolysis reaction mixture included 750 µg of gliadins (MP Biomedicals, Illkirch, France) dissolved in 0.1 M universal acetate–phosphate–borate buffer (UB) [101] at a pH of 6.5, 3 mM DTT, 2 M urea, and *T. molitor* purified enzymes equal to 1.7 larvae. The final volume was 307.5 µL. In total, 20 μL of the reaction mixture was removed at time zero and at 7 min intervals over a 35 min incubation period at 37 °C. The reaction with TNBS was carried out immediately according to the protocol described in [100]. The optical absorbance of the colored reaction product was measured at 405 nm with an ELx808 Absorbance Microplate Reader (BioTek Instruments, Inc., USA).

## Figures and Tables

**Figure 1 ijms-24-00579-f001:**
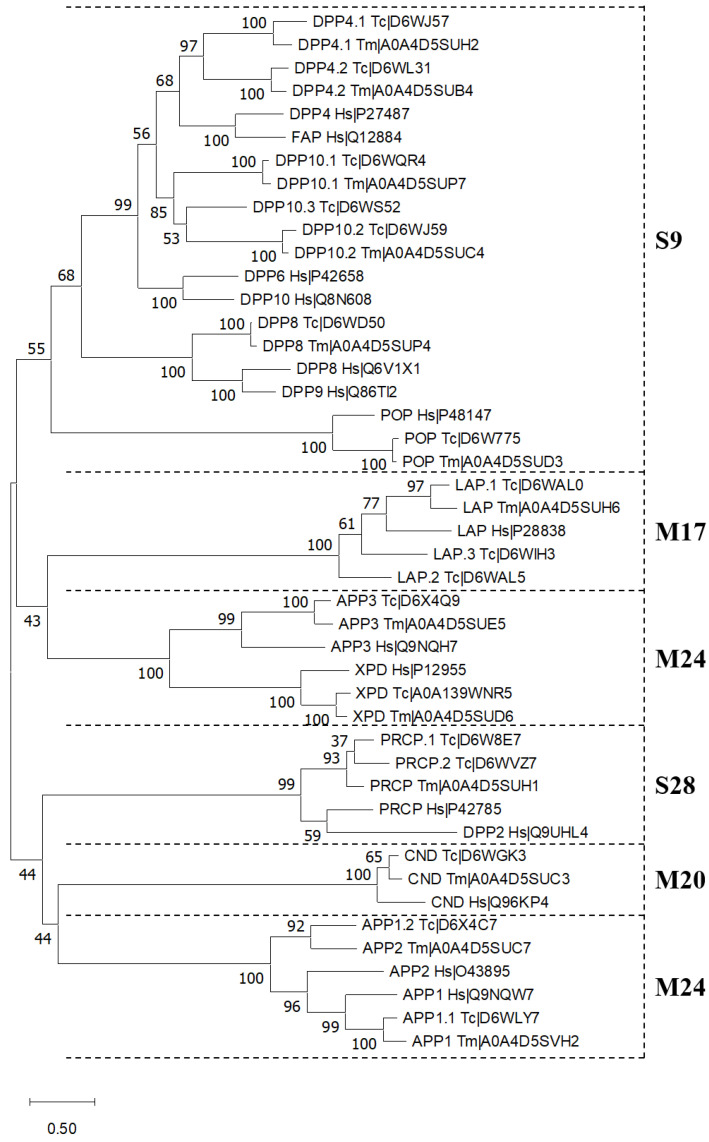
Phylogenetic tree of *T. castaneum*, *T. molitor*, and *H. sapiens* PSPs and PPCPbs constructed using the maximum likelihood method.

**Figure 2 ijms-24-00579-f002:**
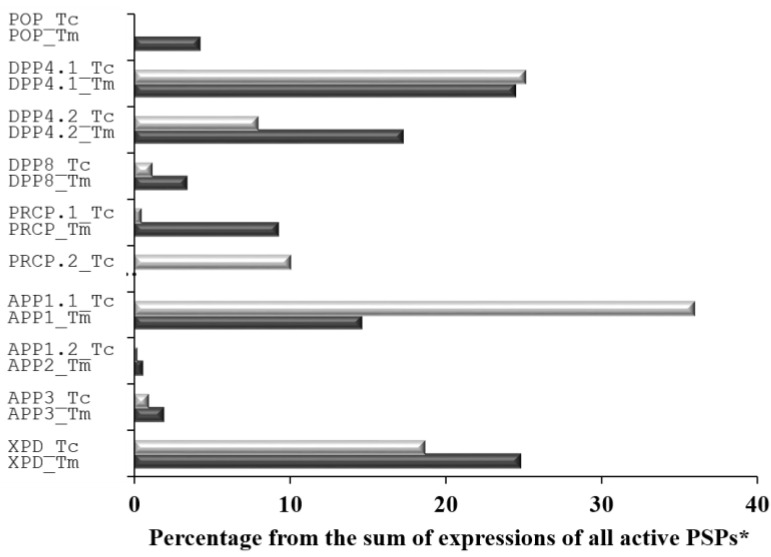
Gut mRNA expression levels of *T. castaneum* (light grey columns) and *T. molitor* (dark grey columns) active PSPs. * This calculation involved only active PSPs: POP, DPP 4, DPP 8, PRCP, APP1, APP2, APP3, and XPD. DPP 10 was not included because it is an inactive DPP 4 homolog and does not possess peptidase activity. LAP and CND were not included because cleavage of post-proline bonds is not their main specificity.

**Figure 3 ijms-24-00579-f003:**
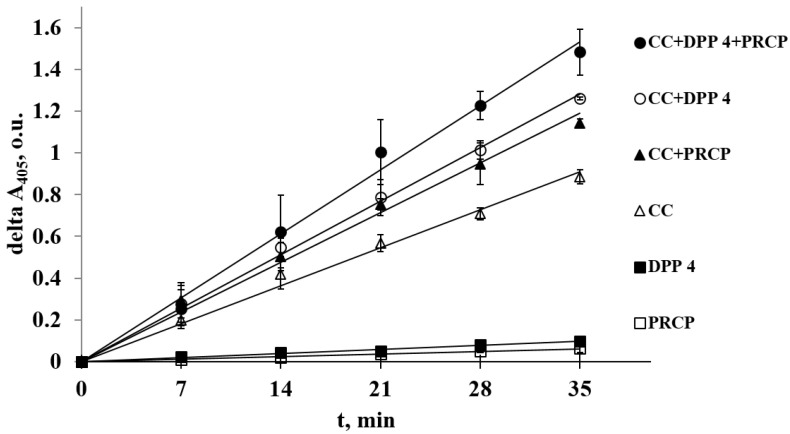
Proteolysis of gliadins by six samples of *T. molitor* digestive peptidase preparations: 1. PRCP (white squares), 2. DPP 4 (black squares), 3. CC (white triangles), 4. CC + PRCP mixture (black triangles), 5. CC + DPP 4 mixture (white circles), and 6. CC + DPP 4 + PRCP mixture (black circles) (description of the curves from bottom to top). Proteolysis of gliadins was detected by an increase in free amino groups using 2,4,6-trinitrobenzene sulfonic acid (TNBS). The error bars represent standard deviation.

**Figure 4 ijms-24-00579-f004:**
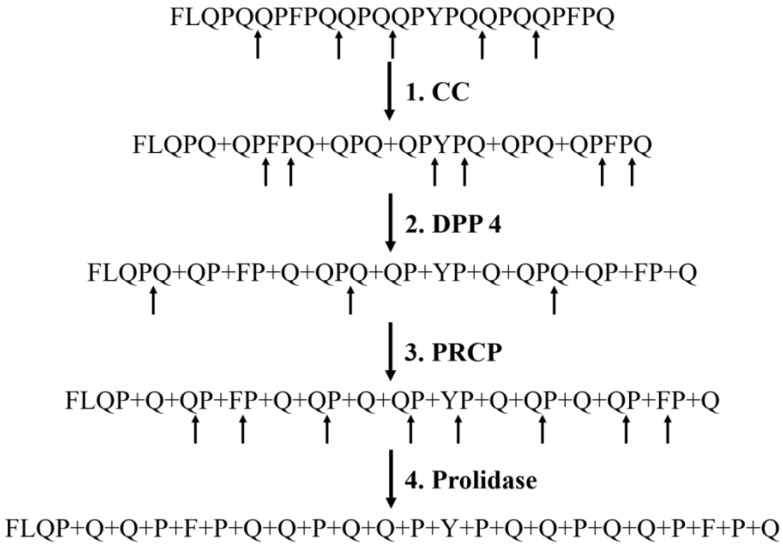
Hypothetical scheme for the complete hydrolysis of glutamine- and proline-rich 26-mer immunogenic gliadin fragments by *T. molitor* and *T. castaneum* digestive CC and PSPs.

**Table 1 ijms-24-00579-t001:** Classification of PSPs found in the *T. castaneum* genome and gut transcriptome and the *T. molitor* gut transcriptome compared with human sequences.

MEROPS Classification	*T. castaneum*			*T. molitor*			*H. sapiens*		
	PSP Name	UniProt ID	NCBI ID	PSP Name	UniProt ID	NCBI ID	PSP Name	UniProt ID	NCBI ID
S9A	POP_Tc *	D6W775 *	XP_015833515 *	POP_Tm	A0A4D5SUD3	QAY29071	POP_Hs	P48147	NP_002717
S9B	DPP4.1_Tc	D6WJ57	XP_975053	DPP4.1_Tm	A0A4D5SUH2	QAY29072	DPP4_Hs	P27487	NP_001926
	DPP4.2_Tc	D6WL31	XP_972016	DPP4.2_Tm	A0A4D5SUB4	QAY29073			
	DPP8_Tc	D6WD50	XP_971949	DPP8_Tm	A0A4D5SUP4	QAY29074	DPP8_Hs	Q6V1X1	NP_932064
							DPP9_Hs	Q86TI2	NP_631898
							FAP_Hs	Q12884	NP_004451
S9X							DPP6_Hs	P42658	NP_570629
	DPP10.1_Tc	D6WQR4	XP_015837625	DPP10.1_Tm	A0A4D5SUP7	QAY29075	DPP10_Hs	Q8N608	NP_065919
	DPP10.2_Tc	D6WJ59	XP_008194406	DPP10.2_Tm	A0A4D5SUC4	QAY29076			
	DPP10.3_Tc *	D6WS52 *	XP_008196030 *						
S28	PRCP.1_Tc	D6W8E7	XP_971305	PRCP_Tm	A0A4D5SUH1	QAY29077	PRCP_Hs	P42785	NP_005031
	PRCP.2_Tc	D6WVZ7	XP_972807						
							DPP2_Hs	Q9UHL4	NP_037511
M24	APP1.1_Tc	D6WLY7	XP_974698	APP1_Tm	A0A4D5SVH2	QAY29078	APP1_Hs	Q9NQW7	NP_001161076
	APP1.2_Tc	D6X4C7	EEZ97287						
				APP2_Tm	A0A4D5SUC7	QAY29079	APP2_Hs	O43895	NP_003390
	APP3_Tc	D6X4Q9	XP_008199099	APP3_Tm	A0A4D5SUE5	QAY29080	APP3_Hs	Q9NQH7	NP_071381
	XPD_Tc	A0A139WNR5	XP_971576	XPD_Tm	A0A4D5SUD6	QAY29081	XPD_Hs	P12955	NP_000276
M17	LAP.1_Tc	D6WAL0	EEZ97983	LAP_Tm	A0A4D5SUH6	QAY29082	LAP_Hs	P28838	NP_056991
	LAP.2_Tc *	D6WAL5 *	XP_974656 *						
	LAP.3_Tc *	D6WIH3 *	XP_972664 *						
M20	CND_Tc	D6WGK3	XP_971451	CND_Tm	A0A4D5SUC3	QAY29083	CND_Hs	Q96KP4	NP_060705.2

* These PSP sequences were found only in the *T. castaneum* genome and were absent in the gut transcriptome. PSP name: the name before the dot means the type of peptidase, the number after the dot can occur if there are several variants of sequences of one type of peptidase, and the abbreviation after the underscore refers to the organism (Tc—*T. castaneum*, Tm—*T. molitor*, Hs—*H. sapiens)*.

**Table 2 ijms-24-00579-t002:** General characteristics of the serine PSP sequences from the S9 and S28 families.

PSP	Sequence Name	Active Site Residues			Residues of S1 Binding Subsite	Residues of S2 Binding Subsite	Residues of S3 Binding Subsite
		Ser	Asp	His			
POP	POP_Hs	S554	D641	H680	W595, F476, V644, V580, Y599, Y473	R643	F173, M235, C255, I591, A594
	POP_Tc	S	D	H	W, F, V, V, Y, Y	R	F, R, C, I, A
	POP_Tm	S	D	H	W, F, V, V, Y, Y	R	F, R, C, I, A
DPP 4	DPP4_Hs	S630	D708	H740	Y631, V656, W659, Y662, Y666, V711	N710, R125, E205, E206	-
	DPP4.1_Tc	S	D	H	Y, V, L, Y, Y, V	N, R, E, E	-
	DPP4.2_Tc	S	D	H	Y, V, W, Y, Y, V	N, R, E, E	-
	DPP4.1_Tm	S	D	H	Y, V, F, Y, Y, V	N, R, E, E	-
	DPP4.2_Tm	S	D	H	Y, V, W, Y, Y, V	N, R, E, E	-
DPP 8	DPP8_Hs	S	D	H	Y, V, W, Y, Y, V	N, -, E, E	-
	DPP8_Tc	S	D	H	Y, V, W, Y, Y, V	N, -, E, E	-
	DPP8_Tm	S	D	H	Y, V, W, Y, Y, V	N, -, E, E	-
DPP 9	DPP9_Hs	S	D	H	Y, V, W, Y, Y, V	N, -, E, E	-
FAP	FAP_Hs	S	D	H	Y, V, W, Y, Y, V	N, R, E, E	-
DPP 6	DPP6_Hs	D	D	H	Y, I, F, Y, F, I,	K, Q, E, E	-
DPP 10	DPP10_Hs	G	D	H	Y, I, L, Y, F, V	K, H, E, E	-
	DPP10.1_Tc	G	D	H	Y, I, W, H, F, V	S, K, N, E	-
	DPP10.2_Tc	G	D	H	Y, I, W, Y, F, A	T, R, E, D	-
	DPP10.3_Tc	S	D	H	Y, V, W, Y, Y, V	N, R, E, E	-
	DPP10.1_Tm	G	D	H	Y, I, W, Y, F, V	S, K, N, E	-
	DPP10.2_Tm	G	D	H	Y, I, W, Y, F, A	T, R, E, D	-
PRCP	PRCP_Hs	S179	D430	H455	M183, W359, M369, W432	-	-
	PRCP.1_Tc	S	D	H	M, W, M, W	-	-
	PRCP.2_Tc	S	D	H	M, W, M, W	-	-
	PRCP_Tm	S	D	H	M, W, M, W	-	-
DPP 2	DPP2_Hs	S	D	H	M, W, L, W		

**Table 3 ijms-24-00579-t003:** General characteristics of the metallodependent PSP sequences from the M24B subfamily.

PSP	Sequence Name	Residues of Mn^2+^-Binding Sites				
		I	I/II	II	II	I/II
APP1	APP1_Hs	D415	D426	H489	E523	E537
	APP1.1_Tc	D	D	H	E	E
	APP1.2_Tc	D	D	H	E	E
	APP1_Tm	D	D	H	E	E
APP2	APP2_Hs	D	D	H	E	E
	APP2_Tm	D	D	H	E	E
APP3	APP3_Hs	D	D	H	E	E
	APP3_Tc	D	D	H	E	E
	APP3_Tm	D	D	H	E	E
XPD	XPD_Hs	D	D	H	E	E
	XPD_Tc	D	D	H	E	E
	XPD_Tm	D	D	H	E	E

**Table 4 ijms-24-00579-t004:** General characteristics of the LAP and CND sequences from the M17 and M20 families.

PPCPbs	Sequence Name	Residues of Zn^2+^-Binding Sites							
		II	I/II		II	I	I/II		
LAP	LAP_Hs	K282	D287	K294	D305	D364	E366	R368	
	LAP.1_Tc	K	D	K	D	D	E	R	
	LAP.2_Tc	K	D	K	D	D	E	R	
	LAP.3_Tc	K	N	K	D	D	A	R	
	LAP_Tm	K	D	K	D	D	E	R	
		**Residues of Mn^2+^-Binding Sites**							
		**II**		**I/II**		**I**	**I/II**		**I**
CND	CND_Hs	H99	D101	D132	E166	E167	D195	H228	H445
	CND_Tc	H	D	D	E	E	D	H	H
	CND_Tm	H	D	D	E	E	D	H	H

**Table 5 ijms-24-00579-t005:** Analysis of the localization domains of PSPs.

PSP	Sequence Name	Length	Protein Mm (Da)	SignalP-6.0 *	TargetP-2.0 *		DeepTMHMM Server	Experimental Data (For Mammalian PSPs Only)
				Signal Peptide	Signal Peptide	Mitochondrial Transit Peptide	TMhelix	
POP	POP_Hs	710	80,700					Cytoplasm [28]
	POP_Tc	704	80,212					
	POP_Tm	705	80,116		1–21 (0.90/0.79)			
DPP 4	DPP4_Hs	766	88,279		1–22 (0.78/0.40)		7–27	Membrane [30,31], soluble [47]
	DPP4.1_Tc	767	87,333	1–19 (0.99/0.98)	1–19 (0.99/0.73)			
	DPP4.2_Tc	821	92,585				34–54	
	DPP4.1_Tm	768	86,576	1–19 (0.99/0.98)	1–19 (0.99/0.78)			
	DPP4.2_Tm	803	90,278				34–54	
DPP 8	DPP8_Hs	898	103,358					Cytoplasm [26,33]
	DPP8_Tc	825	95,048					
	DPP8_Tm	825	94,870					
DPP 10	DPP10_Hs	796	90,888				35–55	Extracellular membrane [26,35]
	DPP10.1_Tc	835	96,053				37–58	
	DPP10.2_Tc	845	95,598				46–66	
	DPP10.3_Tc	858	97,129				25–45	
	DPP10.1_Tm	843	96,808				45–66	
	DPP10.2_Tm	882	99,854				79–100	
PRCP	PRCP_Hs	496	55,800	1–21 (0.99/0.98)	1–21 (0.99/0.48)			Lysosome [36,37,38]
	PRCP.1_Tc	488	55,131	1–16 (0.99/0.98)	1–16 (0.92/0.88)			
	PRCP.2_Tc	478	53,843	1–17 (0.99/0.98)	1–17 (0.99/0.96)			
	PRCP_Tm	488	55,105	1–15 (0.99/0.98)	1–15 (0.99/0.87)			
APP 1	APP1_Hs	623	69,918					Cytoplasm [43]
	APP1.1_Tc	615	69,050					
	APP1.2_Tc	690	79,273	1–13 (0.99/0.98)	1–14 (0.89/0.37)			
	APP1_Tm	partial	partial					
APP 2	APP2_Hs	674	75,625	1–21 (0.99/0.98)	1–23 (0.99/0.54)			Plasma membrane [42]
	APP2_Tm	709	81,041	1–25 (0.83/0.81)	1–25 (0.99/0.97)			
APP 3	APP3_Hs	507	57,034			1–31 (0.89/0.72)		Mitochondrion [44]
	APP3_Tc	503	57,581			1–22 (0.58/0.43)		
	APP3_Tm	504	57,356			–(0.32)		
XPD	XPD_Hs	493	54,548					-
	XPD_Tc	487	54,319					
	XPD_Tm	489	54,168					
LAP	LAP_Hs	519	56,166			1–27 (0.95/0.60)		Microsomes, cytoplasm [45]
	LAP.1_Tc	447	48,087					
	LAP.2_Tc	530	58,105			1–26 (0.94/0.71)		
	LAP.3_Tc	534	58,852			1–21 (0.93/0.81)		
	LAP_Tm	513	55,124			1–24 (0.91/0.31)		
CND	CND_Hs	475	52,878					Cytoplasm [46]
	CND_Tc	478	53,583					
	CND_Tm	478	53,794					

* After the cleavage site position of the signal peptide or mitochondrial transit peptide in brackets, the number before the slash is the likelihood of the specified localization domain appearing and the number after the slash is the probability of cleavage at the indicated position.

**Table 6 ijms-24-00579-t006:** Localization of *T. molitor* larval midgut peptidases *.

	PSPs				
	DPP 4	PRCP	POP	APP	XPD
**Part of the midgut**	**Peptidase activity, % from the total activity (AM + PM)**				
AM	60	73	49	52	37
PM	40	27	51	48	63
	**Tissue distribution, %**				
Soluble contents	81	100	4	18	29
Insoluble contents	3	-	-	5	4
Soluble tissue fraction	7	0	94	58	57
Insoluble tissue fraction	9		2	19	10

* The relative content of each peptidase in different fractions is presented.

**Table 7 ijms-24-00579-t007:** Gut mRNA expression levels of *T. castaneum* and *T. molitor* PSPs and PPCPbs.

Peptidase	*T. castaneum*			*T. molitor*		
	Sequence Name	Expression (RPKM)	Percentage from the Sum of Expressions of All Active PSPs *	Sequence Name	Expression (RPKM)	Percentage from the Sum of Expressions of All Active PSPs *
POP	POP_Tc	-	0	POP_Tm	25	4.2
DPP 4	DPP4.1_Tc	1688	25.1	DPP4.1_Tm	147	24.6
	DPP4.2_Tc	533	7.9	DPP4.2_Tm	102	17.1
DPP 8	DPP8_Tc	75	1.1	DPP8_Tm	20	3.3
DPP 10	DPP10.1_Tc	8	-	DPP10.1_Tm	370	-
	DPP10.2_Tc	5	-	DPP10.2_Tm	7	-
PRCP	PRCP.1_Tc	27	0.4	PRCP_Tm	53	8.9
	PRCP.2_Tc	675	10.0			
APP1	APP1.1_Tc	2420	35.9	APP1_Tm	88	14.7
	APP1.2_Tc	8	0.1			
APP2				APP2_Tm	3	0.5
APP3	APP3_Tc	58	0.9	APP3_Tm	12	2.0
XPD	XPD_Tc	1254	18.6	XPD_Tm	148	24.7
LAP	LAP.1_Tc	86	-	LAP_Tm	15	-
CND	CND_Tc	2462	-	CND_Tm	625	-

* This calculation involved only active PSPs: POP, DPP 4, DPP 8, PRCP, APP1, APP2, APP3, and XPD. DPP 10 was not included because it is an inactive DPP 4 homolog and does not possess peptidase activity. LAP and CND were not included because cleavage of post-proline bonds is not their main specificity.

## Data Availability

Not applicable.

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
