# Peer review of "Complex of Proline-Specific Peptidases in the Genome and Gut Transcriptomes of Tenebrionidae Insects and Their Role in Gliadin Hydrolysis"

_ijms, 2022, doi:10.3390/ijms24010579_

Round 1

Reviewer 1 Report

The paper is well-written and has used standard procedures to determine what digestive enzymes are present in the digestive system of tenebrionid larvae.  They also were able to demonstrate that secreted enzymes predicted from the genetic information were corroborated by actually measuring the digestive system enzymes present in the lumenal contents and the membrane enzymes were in the cell walls.  Overall a good paper even though I am not sure how they plan to deliver these enzymes to patients with the digestive problems.

Author Response

Response to Reviewer 1 Comments,

  1. “but” removed, as suggested.

“Although there are no reliable data on the localization of such peptidases in the studied insects, but there is some information about localization of mammalian peptidases.”

changed to:

“Although there are no reliable data on the localization of such peptidases in the studied insects, there is some information about localization of mammalian peptidases.”

  1. Punctuation removed, as suggested.

“The first stage of hydrolysis was carried out by endopeptidases CС, which can hydrolyze bonds, formed by glutamine residues.”

changed to:

“The first stage of hydrolysis was carried out by endopeptidases CС, which can hydrolyze bonds formed by glutamine residues.”

  1. “Those” was clarified: the sequences for which the annotation has been refined are listed in brackets.

“Those of the found sequences, which lacked annotation in the database, were annotated during the subsequent construction of a phylogenetic tree and analysis of multiple alignments.”

changed to:

“Those of the found sequences, which lacked annotation in the database (namely, DPP4.2_Tc, DPP10.2_Tc, DPP10.3_Tc), were annotated during the subsequent construction of a phylogenetic tree and analysis of multiple alignments.”

Reviewer 2 Report

Tereshchenkova et al. report the (presumably complete) set of proline-specific digestive proteases from two insect pests that consume grain as their sole food source. Since these insects depend on plant storage protein (gliadins) as an amino acid source for protein synthesis as well as an energy source, it stands to reason that they have the capability to digest it in a manner that cannot be accomplished by humans, where the indigestible components of gliadins contribute to celiac disease. Gliadins are unusual proteins with an exceptionally high proportion of proline residues, which are resistant to hydrolysis by the principal digestive enzymes. They identify the set of genes from the insect genomes, characterize the signal sequences and trans-membrane domains, and probable cellular locations using standard and appropriate bioinformatics tools. They analyze the expression of proline-active enzyme transcripts in the midgut of both insects with RNAseq (reported in https://www.sciencedirect.com/science/article/pii/S2352340921005850) and measure the activities using enzymatic assays.

The research group has significant experience with insect digestive enzymes and particularly gliadin digestion in the family Tenebrionidae.  

With their whole-genome analysis, it is likely that they have found the entire suite of enzymes involved in digesting gliadins. Since they have access to the respective beetle genomes, it should be simple to report the genomic organization of these genes with respect to each other. Are they in the same linkage group?  Some additional context on the genomic organization of the genes with respect to the chromosomes could be relevant to the regulation of these genes, and would not require any additional experiments.

Since the group has already published several papers on proline-active digestive enzymes, they could be more clear about what is novel in this paper and how this particular manuscript adds information that was not in any of their previously published papers. For example, Tereshchenkova et al. (2016) had already characterized DPP4 expression in the anterior vs. posterior midgut of T. molitor, but this was not mentioned in line 401 They had already done the gene structural analysis to show that it was secreted. I think it was a matter of writing rather than anything, but the way this is presented, this would appear to be a novel finding of the current manuscript.

The current manuscript needs to be more clear about how unlike the previous manuscript, where they only used DPP4, in this manuscript, they analyzed combinations of enzymes to assess whether they could achieve a higher level of degradation of gliadins.

https://www.sciencedirect.com/science/article/am/pii/S0965174816300923

Table 6 seems to be the result of a single experiment with no replicates. If not, then some indication of error should be mentioned.

Figure  3     Please indicate in the figure legend the number of samples and what the error bars represent (SEM?). Legend would be more helpful if it explained the readout of the assay (detection of an increase of free amino groups with a colorimetric substrate).  It shouldn’t be necessary to refer to the methods section to see what the graph is referring to. 

The results of this paper could contribute to the design of a cocktail of enzymes that could aid in digesting gliadins in the human gut, which would be an advance in the treatment of celiac disease.  Though it could be streamlined, the paper contributes some new information to this field.

Specific Suggestions

Line 12   proline-specific   (change throughout)

Line 14  PSP sequences

Line 25  .. These results show promise that a drug for the enzymatic therapy of celiac disease and gluten intolerance based on tenebrionid digestive enzymes could be developed.

Line 30  in the family Tenebrionidae

Line 35  ..and the only imino…

Line 38  … provides a rigid…

Line 40  gliadin-containing

Line 48   This, the study of..

Line 79   Among the human intestinal PSPs that could participate in digestion, only DPP4 [11] and XPD [12-15] activities have been reported in the literature.

Line 84    in the human genome.

Line 102   using the larval

Line 107  human and insect…

Line 110     insects, unlike humans, had some

Line 122    highly conserved

Line 132   from the S9B subfamily

Line 133  The active site…

Line 139  Active insect peptidases  (change insects’ throughout)

Line 182    The phylogenetic tree

Line 239  mitochondrial

Line 277  The expression…

Line 326   capable of cleaving

Line 373   A bioinformatic…

Line 427 capable of participating…

Line 430  Gliadin hydrolysis

Line 447  The suggested…

Line 456  This indicates that it may be possible to develop an enzyme-therapy drug for celiac disease and gluten intolerance based on tenebrionid digestive enzymes.   

Line 560    peptidases, the extract

Line 570  proteolysis of gliadins

Author Response

Response to Reviewer 2 Comments,

  1. “With their whole-genome analysis, it is likely that they have found the entire suite of enzymes involved in digesting gliadins. Since they have access to the respective beetle genomes, it should be simple to report the genomic organization of these genes with respect to each other. Are they in the same linkage group?  Some additional context on the genomic organization of the genes with respect to the chromosomes could be relevant to the regulation of these genes, and would not require any additional experiments.”

Thank you for your suggestion. In this paper we would like to focus on the analysis of transcriptomic data mostly because this allows to propose the importance of the studied PSPs in accordance with expression levels evaluation. The genomic organization of the genes in tenebrionids will be included in our next separate paper.

  1. “Since the group has already published several papers on proline-active digestive enzymes, they could be more clear about what is novel in this paper and how this particular manuscript adds information that was not in any of their previously published papers. For example, Tereshchenkova et al. (2016) had already characterized DPP4 expression in the anterior vs. posterior midgut of T. molitor, but this was not mentioned in line 401 They had already done the gene structural analysis to show that it was secreted. I think it was a matter of writing rather than anything, but the way this is presented, this would appear to be a novel finding of the current manuscript.”

Thank you for your comment. We referred to the relevant articles in the Results section, but now, according to your remark, we have duplicated them in the Discussion section too. Now the relevant paragraph (in line 401) has been rewritten as follows:

“Using our previous and new data on localization of T. molitor PSPs we can summarize that DPP 4 [48] and PRCP [29] were located mainly in the AM, POP [29] and newly studied APP were distributed evenly between AM and PM parts of the gut, and XPD [49] was localized predominantly in PM. Analysis of tissue distribution of the PSPs showed that DPP 4 [48] and PRCP [29] were located in soluble gut contents and can participate in T. molitor digestion, and this is in agreement with bioinformatic data ‑ these peptidases were predicted to be secreted and highly expressed.”

  1. “The current manuscript needs to be more clear about how unlike the previous manuscript, where they only used DPP4, in this manuscript, they analyzed combinations of enzymes to assess whether they could achieve a higher level of degradation of gliadins.

https://www.sciencedirect.com/science/article/am/pii/S0965174816300923”

We added the information about previous studies to the paragraph (in line 428):

“Previously we have shown that T. molitor DPP 4 is able to hydrolyze gliadins more efficiently than human DPP 4 [48]. In the present study, we tested the natural complex of digestive enzymes of tenebrionids, providing efficient hydrolysis of their proline-rich natural food substrates like gliadins in natural conditions and ensuring sufficient intake of proline, to find whether this would increase the efficiency of hydrolysis.”

  1. “Table 6 seems to be the result of a single experiment with no replicates. If not, then some indication of error should be mentioned.”

In the localization studies activity was determined by special substrates in the AM and PM parts of the gut in 3 biological replicates and 3-5 technical repeats. The standard deviation did not exceed 5-10%. Then the relative content of each peptidase in the fractions was calculated for the convenience of analysis and this data are presented in the Table 6.

We added this additional information to the section 4.9. in Methods and explained in the footnote to the table 6 that it presents not experimental, but calculated data for which errors are not used.

  1. “Figure  3     Please indicate in the figure legend the number of samples and what the error bars represent (SEM?). Legend would be more helpful if it explained the readout of the assay (detection of an increase of free amino groups with a colorimetric substrate).  It shouldn’t be necessary to refer to the methods section to see what the graph is referring to.”

We have added the details to the legend and the caption to the Figure 3:

“Proteolysis of gliadins by six samples of T. molitor digestive peptidases preparations: 1. PRCP (white squares), 2. DPP 4 (black squares), 3. CС (white triangles), 4. CС + PRCP mixture (black triangles), 5. CС + DPP 4 mixture (white circles), and 6. CС + DPP 4 + PRCP mixture (black circles) (description of the curves from bottom to top). Proteolysis of gliadins was detected by the increase of free amino groups using 2,4,6-trinitrobenzene sulfonic acid (TNBS). The error bars represent standard deviation.”

Specific Suggestions.

All specific suggestions were thoroughly corrected.
